# The Effect of a Company's Sustainable Competitive Advantage on Their Tax Avoidance Strategy—Focusing on Market Competition in Korea

Yoojin Shin [1] and Jung-Mi Park [2,*]

1. Division of Business, Chosun University, Gwangju 61452, Republic of Korea; yoojinshin@chosun.ac.kr
2. Division of Business Administration, Wonkwang University, Iksan 54538, Republic of Korea
* Correspondence: mafi2050@wku.ac.kr; Tel.: +82-63-850-5768

**Abstract:** This study analyzes whether a company's competitive advantage affects a company's tax avoidance strategy. Additionally, it analyzes whether these effects depend on the level of competition in the market to which the company belongs. This is because a company's tax avoidance strategy may vary depending on the characteristics of the firm, such as the financial position and governance structure, the market dominance, or the degree of competition in the market to which the company belongs and it can act as an incentive for tax avoidance. Results of this study is follows. Tax avoidance increases significantly as a company's market share increases. Also, if the sample is divided by the level of market competition and analyzed, the results show that tax avoidance increases significantly with the increase in a company's market power only in oligopolistic markets with low market competition. Therefore, it can be interpreted that the effect of a company's market power on tax avoidance varies depending on the level of competition in the market.

**Keywords:** tax avoidance; tax strategies; market competition

## 1. Introduction

This study aims to analyze the effect of a company's sustainable competitive advantage on its tax avoidance strategy and examines whether the effect differs according to the level of competition in its relevant market. A company's tax avoidance strategies may differ depending on its financial status or administration, among other characteristics, and its market power or the level of competition in its relevant market may act as an incentive for tax avoidance. However, in many prior studies, only the effect of individual companies' competitive advantage on tax avoidance or the effect of market competition on tax avoidance was analyzed. However, depending on the level of market competition, the effect of a company's competitive advantage on tax strategies is expected to differ. Therefore, we analyze the impact of a company's sustainable competitive advantage on its tax avoidance strategy and examine whether this effect differs depending on the level of competition in its belonging market.

Tax avoidance benefits companies by increasing their available cash in the short term, but it may reduce their sustainability in the long term. Tax avoidance may result in future cash outflows due to tax collections and increased tax volatility and tax risk [1]. Hence, tax avoidance negatively affects the sustainability of a company. Dhawan et al. [2] reported that tax avoidance directly influences the bankruptcy risk of a company, and Lee and Kim [3] found that companies that prioritize sustainable management are less likely to avoid taxes.

Top management teams (TMTs) have the strongest authority for a company's strategic choices, their incentives for tax avoidance vary according to their management strategies and business environment. According to the perspective theory, companies set different levels of risk they can take depending on their performance. Tax avoidance is closely related to corporate risk. It is determined by corporate tax strategies, and influenced by

executive compensation and governance, access to capital, and the financial shape of the company. Companies with weak external monitoring or vulnerable governance are more likely to utilize tax avoidance strategies. Recent studies have revealed that the competitive corporate environment may act as a governance structure, and a company's sustainable competitive advantage or the level of competition in its relevant market can become an incentive for tax avoidance [4–6].

The market power of a company refers to its competitive advantage in its belonging market, and it is often measured by its market share [7]. A high market power implies that a company has significant pricing power in its market and can gain a competitive advantage in the market relatively easily. Therefore, a company's market power affects its managerial decisions [8]. In other words, companies with different levels of market power play different roles in the market, follow different management strategies, and make different managerial decisions [9]. Consequently, market power can influence their chosen tax avoidance strategy. Previous studies have found that companies with greater market power are more likely to perform aggressive tax avoidance [4,10].

A company's market power can differ depending on the characteristics of its industry and the level of competition in its relevant market, which influences not only the company's disclosure policy and earnings quality but also its agency problem, for better or for worse [11–13]. These differences also affect a company's internal financing, which can act as an incentive for tax avoidance. Since tax avoidance reduces cash outflow, it can be used for internal financing [14]. To sum up, not only the market power of a company but also the level of competition in its relevant market is expected to affect the company's tax avoidance strategy. However, it is difficult to find previous studies that have studied their relationship in connection with each other.

Against this backdrop, we aim to analyze the effect of a company's sustainable competitive advantage on its tax avoidance strategy and examine whether this effect differs according to the level of competition in its relevant market. Even companies with the same level of market power are expected to have different competitive positions based on the market's level of competitiveness. In other words, a company with high market power in a market with a low level of competition would be a monopolist, whereas a company with high market power in a perfectly competitive market with a high level of competition is not. Therefore, as a company's competitive position will affect its tax avoidance strategy differently according to the level of market competition, it should be considered in analysis. In this study, we study whether the impact on a company's tax avoidance—assuming the company has strong market power—varies in each of the three markets: perfectly competitive, moderately competitive, and oligopolistic markets.

Previous studies have only analyzed the effect of individual companies' competitive advantage on tax avoidance or the effect of the level of competition in the market to which the company belongs on that company's tax avoidance. However, depending on the level of market competition, a company's competitive advantage can not only affect the company's sustainability differently, but also the strategy that the company should take. From this point of view, this study differs from previous studies in that it provides empirical analysis results that take into account not only the competitive position of the company but also the level of competition in the market to which the company belongs. The remainder of the paper is organized as follows. Section 2 reviews previous research and examines the research hypotheses. The research model is designed in Section 3. Section 4 presents the descriptive statistics and results of empirical analysis of the correlations and research model. Section 5 presents the summary and conclusion of this study.

## 2. Literature Review and Research Design

### 2.1. Literature about Tax Avoidance

Business enterprises utilize tax avoidance strategies to reduce tax liability. Tax avoidance, an action taken to avoid tax burden, is a broad concept that includes tax evasion and tax saving. Empirical research on taxation is typically categorized into tax avoidance, tax

aggressiveness, tax sheltering, and tax evasion [15]. Among these concepts, the scope of tax avoidance is relatively broad, as per its definition as a series of corporate activities to reduce tax liabilities [16].

Corporate tax avoidance activities are strategically determined. By minimizing tax burden, a company can not only increase its cash reserves but also use them for internal financing. Also, investment based on this can increase the wealth of the shareholders [17]. However, if tax avoidance were to be detected, it could cause the incurrence of additional taxes and increase cash outflows owing to additional tax liabilities, thus leading to future tax risk [16,18,19].

Tax avoidance could pose the future risk of a company. Badertscher et al. [20] mentioned that tax avoidance is a dangerous activity that may result in future risk and the incurrence of substantial cost in the future, while Dyreng et al. [16] stated that tax avoidance increases future tax risk. In other words, tax avoidance increases tax uncertainty, which may be related to a company's future risk. In line with this view, previous studies have analyzed the direct impact of tax avoidance on a company's future risk. For example, Rego and Wilson [21] analyzed the relationship between stock price risk and tax risk and found a significant positive relationship between tax avoidance and corporate risk. Balakrishnan et al. [22] analyzed whether companies with aggressive tax avoidance have a high level of information opacity. Corporate tax avoidance may increase a company's cash reserves by reducing the tax burden, but as it cannot be known form outside, information asymmetry is created. Consequently, aggressive tax avoidance lowers corporate transparency. Lastly, Kim et al. [23] analyzed the effect of tax avoidance on the risk for a stock market crash. They argued that tax avoidance can lead to opportunistic behavior by managers and that the accumulation of these effects and their sudden disclosure to the market could cause a sudden dramatic decline in stock prices. The analysis revealed that the risk for a stock price crash increases with an increase in the practice of tax avoidance. Dhawan et al. [2] found tax avoidance to have a direct effect on the risk of corporate bankruptcy, while Lee and Kim [3] found that companies that engage more in sustainable management are less likely to avoid taxes. To sum up, a company's tax avoidance strategy is highly important because it influences its risk and sustainability.

As a business strategy, there are various incentives for tax avoidance, as discussed in many previous studies [24–26]. Armstrong et al. [24] checked whether executive compensation acts as an incentive for tax avoidance and reported that it plays a role in lowering the effective tax rate. Brown et al. [25] analyzed the effect of executive compensation on tax avoidance as well and reported that tax avoidance increased as executive compensation increased, while executive compensation decreased when tax risk increased. Desai and Dharmapala [26] examined whether stock compensation for executives acted as an incentive for tax avoidance and found that it had a significant negative effect on tax avoidance. They reported that tax avoidance decreases as stock compensation for executives increases, particularly in situations with weak governance.

In addition to executive compensation, governance is an important factor that determines tax avoidance. Chen et al. [27] examined whether corporate ownership, gauged in terms of whether a company is family-owned, influences tax avoidance. They found that family companies use less aggressive tax avoidance strategies than do non-family companies. They interpreted the findings as suggesting that managers of family companies are willing to forgo the benefits of tax avoidance to avoid the non-tax costs of tax avoidance. Minnick and Noga [28] examined the effect of board independence on tax avoidance and found that greater board independence is associated with more effective tax avoidance owing to empirical knowledge, and this effect is more pronounced with a smaller board of directors (BOD). These results suggest that the role of the board as a governance structure can influence a company's tax avoidance strategy.

In addition, the level of tax avoidance can be determined by a company's access to capital, accounting choice, or business environment. Notably, when information asymmetry exists in a company, the cost of external financing is likely to increase or the possibility

of financing is likely to decrease. In such cases, companies have an incentive to utilize funds generated internally through tax avoidance rather than depending on external financing [14,29]. Edwards et al. [14] examined whether internal financing is increased by reducing cash outflows through tax avoidance in the presence of financial constraints. They found that as financial constraints increased, companies increased internal financing through tax planning.

Watson [30] analyzed the relationship between corporate social responsibility (CSR) and tax avoidance. This study reported that companies that engage in fewer CSR activities employ more tax avoidance strategies. This relationship was found to be stronger when the company's performance was low and weaker when the company's performance was high. Gallemore and Labro [31] also analyzed whether a company's internal information environment can provide incentives for tax avoidance and revealed that higher internal information quality is associated with increased tax avoidance (and lower effective tax rates). Lastly, Choi et al. [32] analyzed the impact of related party transactions on tax avoidance and found that tax avoidance increases with the size of the related party transaction.

Overall, it can be interpreted that a company's tax avoidance increases in situations with weak external monitoring or governance. Consequently, tax avoidance strategies may vary depending on the environment in which a company operates; a growing body of research suggests that a company's market power or the level of competition in the market may be an incentive for tax avoidance [4–6].

*2.2. Market Power and Tax Avoidance*

The market power of a company represents its competitive position in a given market. This is determined by the corporate effort to maximize earnings [7], and a company with strong market power plays a role in determining the demand and supply of the market. In other words, high market power means that the company has significant pricing power in the market and can gain a competitive advantage in the market relatively easily.

A company's market power determines the role it plays in the market and the management strategies it can adopt, so its managerial decisions vary depending on its market power [9]. Therefore, the effect of market power on companies needs to be examined. Previous studies have reported that companies with greater market power are more productive and efficient [33]. Further, Leibenstein's [34] X-inefficiency theory shows that the difference between maximum possible output and actual output decreases with greater market power when labor is fully utilized. This means that companies with greater market power are more efficient owing to the increased monitoring by external governance.

As discussed above, the market power of a company induces an increase in corporate productivity and efficiency, thus improving its financial performance and earnings quality. Kale and Loon [35] analyzed whether companies with higher market power have more pricing power in the market, which leads to higher stock liquidity. The results of these studies show that market power reduces the volatility of earnings, so companies with higher market power have more stable cash flows and higher stock liquidity. Mitra et al. [36] analyzed the relationship between market power and earnings management. They found that real-activity-based earnings management is significantly lower in companies with higher market power. Datta et al. [37] examined the relationship between market power and earnings management as well and found that companies with lower market power are more likely to manage earnings through discretionary accruals. This can be interpreted as fewer transparent financial statements and less useful information. This phenomenon is more pronounced in more competitive industries. This suggests that not only the market power of a company but also the level of competition in its industry can affect managers' accounting choices.

A company's market power can also affect tax avoidance. Akdogu and Mackay [10] reported that companies with higher market power are more likely to engage in relatively aggressive tax strategies; this is because such companies are more capable of hedging against risk than other companies owing to the flexibility or predictability of their earnings.

Kubick et al. [4] found that greater market power is associated with greater tax avoidance and reported that non-leading companies tend to follow this strategy in turn when leading companies adopt it.

### 2.3. Market Competition and Tax Avoidance

A leading company is a corporation that currently dominates its industry. The pricing and financial decisions of leading companies vary depending on the level of their market power [4,38–40]. However, the advantages of their strong market power can differ according to the differences in the level of monitoring between companies. For example, in highly competitive, non-monopolistic markets, there would be a high level of monitoring between leading and following companies. Leaders would try to maintain their competitive advantage to retain the benefits associated with market power, while followers would make efforts to regain a competitive advantage because their market power is low. However, in monopolistic or oligopolistic markets, the more market power a company has, the lower the level of monitoring between leaders and followers. This is because monopolistic or oligopolistic markets have fewer non-leaders, and non-leaders are more likely to focus on maintaining or increasing their sustainability to remain afloat.

A company's market power can change according to the nature of its industry and the level of competition in its relevant market. Level of competition refers to three types of markets: perfectly competitive markets, markets with moderate levels of competition, and monopolistic markets with low levels of competition [41]. Perfectly competitive markets are the most competitive type of market, and the degree of competition in the market affects the decision-making of managers [8]. In other words, the degree of market competition can affect a company's disclosure policy or earnings quality [11], which can exacerbate or mitigate agency problems [13,41,42].

The level of competition in a market affects a company's management strategy and quality of accounting information. Ryu et al. [43] analyzed the relationship between the level of competition in industrial markets and earnings management and found an inverse relationship between the two. In other words, the quality of earnings depends on the level of competition in the market. Lee and Shin [44] found that accounting information becomes more comparable with an increase in the level of intra-industry competition. Valta [45] analyzed whether the cost for capital changes with increasing competition in the industry and concluded that the cost of borrowing capital from banks increases as competition increases. It was interpreted that in such cases, managers have an incentive to raise internal funds to generate cash flows because external financing is difficult.

As the competition grows in the market, a company can resort to tax avoidance to secure funding more easily or enhance sustainability. Companies with strong market power are less likely to be affected by competitiveness, which increases their likelihood of risk taking [17]. From this perspective, some researchers, including Shin and Park [5], have empirically analyzed the effect of the level of market competition on a company's tax avoidance. Their analysis showed a significant negative relationship between the level of competition in the market and tax avoidance, and this relationship was stronger with better corporate governance. This suggests that tax avoidance increases as the level of competition in a market decreases, and it can be curbed by good corporate governance. Kim and Lee [6] also analyzed the effect of competitive threat in the market on tax avoidance. They reported that tax avoidance increases with the presence of threats to market competition, and that this effect is stronger in situations with weak governance and low financial flexibility. Lastly, Karamshahi et al. [46] examined the relationship between market competition and tax avoidance. They revealed that the Herfindahl's Index (HHI) and the entry barrier ratio significantly influence tax avoidance. This is interpreted that the more competitive the market, the more tax avoidance occurs. The main topics of the preceding studies presented above are summarized in Table 1 below.

**Table 1.** Summarized prior research.

| Topic | Reference |
|---|---|
| The Effect of Tax Avoidance on the Risk of Companies | Badertscher et al. [20], Rego and Wilson [21], Balakrishnan et al. [22] |
| Incentives for tax avoidance | Armstrong et al. [24], Chen et al. [27], Edwards et al. [14] |
| The Effect of a Company's Market power on Tax Avoidance | Akdogu and Mackay [10], Kubick et al. [4] |
| The Effect of Market Competition Level on Tax Avoidance | Shin and Park [5], Kim and Lee [6] |

*2.4. Research Design*

Akdogu and Mackay [10] and Kubick et al. [4] analyzed whether the higher the market power of individual companies, the more aggressive the tax avoidance. Whereas Shin and Park [5] and Kim and Lee [6] analyzed the effect of competition in the market belonging firms on tax avoidance and reported that the level of tax avoidance varies depending on the degree of competition in the market. However, the market power of individual companies and the degree of competition in the product market have an interactive relationship with each other.

In other words, the market power of a company and the level of market competition, respectively or in interactively, are expected to have a significant effect on the company's tax avoidance strategy. In particular, leading companies have an incentive for tax avoidance to maintain their competitive advantage with stronger market power, and this phenomenon is expected to vary according to the level of market competition. However, previous studies have either analyzed the effect of the market power of a company on tax avoidance or the effect of market competition level on tax avoidance, and few studies have considered both. While previous studies have only analyzed the effect of individual companies' competitive advantages on tax avoidance, this study is different from other studies in that it conducts an empirical analysis considering the level of competition in the market. Therefore, this study proposes the following hypotheses:

**Hypothesis 1.** *The level of tax avoidance varies as a company's market power increases.*

**Hypothesis 2.** *The level of competition in the market influences the effect of a company's market power on tax avoidance.*

**3. Sample and Research Model**

*3.1. Sample*

This study aims to analyze the effect of a company's sustainable competitive advantage on its tax avoidance strategy using non-financial insurance companies listed on the KOSPI and KOSDAQ from 2011 to 2018. Further, we investigate whether this effect differs according to the level of competition in the company's relevant market. The financial data required for the analysis were extracted from the Kis-Value database and the TS-2000 database. The sample was selected in the following manner. First, we excluded companies-years that are not December fiscal year-end entities and do not have the necessary financial data for the analysis. In addition, since it is important to measure the level of competition in the market, we excluded companies-years for the level of market competition cannot be measured. The final sample comprised 4542 company-years. Detailed sample selection procedures are shown in Table 2 below.

**Table 2.** Sampling process.

| Criteria | # of Samples |
|---|---|
| non-financial and insurance companies listed on the KOSPI and KOSDAQ from 2011 to 2018 | 16,728 |
| Fewer firm-years not December at end of accounting period | (878) |
| Fewer firm-years without financial data | (6508) |
| Fewer firm-years without data for measuring market competition and tax avoidance | (4800) |
| Final samples | 4542 |

Table 3 shows the distribution of the sample by year. In 2011, the sample comprised 421 company-years, while in 2018 it comprised 644 company-years. Overall, there is an increasing trend in the number of sample companies from 2011 to 2018. This seems to be due to the increase in the total number of listed companies in the Korean stock market.

**Table 3.** Distribution of the Sample by year.

| Year | No. of Total Samples | Ratio | Low Competition Samples | Mid Competition Samples | High Competition Samples |
|---|---|---|---|---|---|
| 2011 | 421 | 9.27% | 226 | 114 | 81 |
| 2012 | 511 | 11.25% | 297 | 124 | 90 |
| 2013 | 522 | 11.49% | 305 | 91 | 126 |
| 2014 | 563 | 12.40% | 252 | 186 | 125 |
| 2015 | 619 | 13.63% | 266 | 208 | 145 |
| 2016 | 647 | 14.24% | 299 | 202 | 146 |
| 2017 | 615 | 13.54% | 268 | 171 | 176 |
| 2018 | 644 | 14.18% | 281 | 165 | 198 |
| Total | 4542 | 100.00% | 2194 | 1261 | 1087 |

### 3.2. Variables and Research Model

This study not only analyzes the effect of a company's market power on tax avoidance but also investigates whether this effect differs depending on the level of competition in the company's market. Hence, it was essential to measure the market power of firms and the level of market competition. First, market power was measured using revenue shares, an approach consistent with most of the previous studies. Revenue share is defined as an individual company's sales revenue divided by the total revenue of the company's relevant industry, and it represents the proportion in the total revenue of the company's relevant market.

In the literature, most studies use industrial concentration to measure the level of market competition. Grullon and Michaely [41] reported that the level of market competition is inversely correlated with industrial concentration (HHI). Specifically, industrial concentration is the sum of the squared market shares of all companies within a given industry, and market share is measured using sales revenue. It can be expressed by the following formula [47].

$$HHI = \sum_{i=1}^{n} S_{i,j,t} \tag{1}$$

$S_{i,j,t}$: Sales of company $i$ in industry $j$ at time $t$/Total sales of industry $j$ at time $t$.

Here, $S$ is the market share of the industry, measured by revenue. Therefore, its squared value, industrial concentration, is large when a leading company has a high market share among the companies in the industry. This means that if the level of market competition is high and the difference in market share between individual companies is small, industrial concentration (HHI) is small, but if the level of market competition is low

and a monopoly exists, industrial concentration is large. Previous study measured the level of market competition by multiplying the industrial concentration by −1, and some classified the market using the calculated industrial concentration [48]. Sys [48] determined market competitiveness using the value of HHI (industrial concentration) multiplied by 10,000. If the value is less than or equal to 1000, it is classified as a competitive market with low industrial concentration; if the value is between 1000 and 1800, it is classified as a market with moderate competition; and if the value is more than 1800, it is classified as an oligopolistic market. These criteria were applied in this study to classify the market structure into three categories according to the level of market competition.

Another core variable that was used to prove the hypotheses of this study is tax avoidance, which was measured by the effective tax rate (GAAP_ETR) and the cash effective tax rate (CASH_ETR) in previous studies [14,16,19]. The literature shows that measures of tax avoidance vary considerably, but effective tax rate is often used as the primary measure of tax avoidance [1,49,50]. This is because companies use tax avoidance to lower the effective tax rate. The effective tax rate can be broadly categorized into the financial statement effective tax rate and the cash effective tax rate. The effective tax rate is calculated by dividing the current income tax expense by the net income before taxes; the higher the value, the higher the effective tax rate, which indicates no tax avoidance. In addition, the cash effective tax rate is calculated by dividing cash income taxes paid by net income before taxes; the higher the value, the higher the cash effective tax rate, which indicates no tax avoidance [21,51]. Since the effective tax rate and cash effective tax rate variables are inversely related to tax avoidance, we tested our hypotheses by multiplying the effective tax rate and cash effective tax rate variables by −1 to facilitate the interpretation of our results.

The analytical models for H1, which states that the level of tax avoidance varies as a company's market power increases, and H2, which states that the level of competition in the market influences the effect of a company's market power on tax avoidance, are as follows [16,19].

$$
\begin{aligned}
TaxAvoidance = \ &\beta_0 + \beta_1 MKTshare + \beta_2 Size + \beta_3 Leverage \\
&+ \beta_4 CurrRatio + \beta_5 MTB + \beta_6 lnAGE + \beta_7 ROA \\
&+ \beta_8 LossDummy + \beta_9 MKT \\
&+ YearDummy + IndDummy + \varepsilon
\end{aligned}
\tag{2}
$$

| | | |
|---|---|---|
| *Tax Avoidance* | : | Tax Avoidance measured by Cash_ETR, GAAP_ETR |
| *Cash_ETR* | : | cash tax paid divided by before-tax book income |
| *GAAP_ETR* | : | total income tax expense divided by before-tax book income |
| *MKTshare* | : | sales of firm divided by the total sales of the belonging industry |
| *Size* | : | natural logarithm of total assets |
| *Leverage* | : | total liabilities divided by total assets |
| *CurrRatio* | : | current assets divided by underlying assets |
| *MTB* | : | common stock divided by total equity |
| *lnAGE* | : | natural logarithm of current year minus the year of establishment |
| *ROA* | : | net income divided by total assets |
| *Lossdummy* | : | dummy variable that is 1 if loss occurred in the previous year, or 0 otherwise |
| *MTK* | : | market dummy |
| *IndDummy* | : | industry dummies |
| *YearDummy* | : | year dummies |

The model shown above, with tax avoidance as the dependent variable and market power as the main variable of interest, was used to test both H1 and H2. To be specific, we estimated the model for the entire sample to test H1 and analyzed each subsample according to the level of market competition to test H2. If the level of tax avoidance changed as a company's market power increased, consistent with H1, the coefficients of the cash effective tax rate and effective tax rate variables in the model were expected to be significantly positive or negative. Additionally, if this effect depends on the level of

competition in the market, there would be significant differences in the coefficients of the cash effective tax rate and effective tax rate variables in the subsamples analyzed by the level of market competition.

The control variables included in the model are as follows [16,19]. First, financial condition, such as company size, debt ratio, and current ratio, is a representative factor that affects the tax avoidance of companies, and it was set as a control variable in the above model. In addition, the age of the company, return on assets, and the ratio of a stock's market value to its book value were included because the growth of the company may also affect its tax avoidance strategy. In addition, the loss dummy variable was included because loss-making companies may be incentivized to avoid taxes. Finally, we included market, industry, and year dummies in the model. This study attempted to solve omitted variable problems by using the included control variable in-model, but still the possibility of the omitted variable may exist. In this study, the tax avoidance measure was measured at the effective tax rate, but it also would be meaningful to analyze it using other tax avoidance measures such as BTD. However, the model (2) of this study has not been studied in previous studies, and this is the part where this study differs from other studies. By analyzing this and presenting the results, this study will contribute to related topics.

## 4. Empirical Analysis Result

### 4.1. Descriptive Statistics

Table 4 presents the descriptive statistics of the variables used in the analysis. First, the descriptive statistics for the full sample show that the mean (median) cash effective tax rate (CASH_ETR) is −0.223 (−0.211) and the mean (median) effective tax rate (GAAP_ETR) is −0.222 (−0.211). In addition, the mean (median) market share (MKTShare) is 0.045 (0.010). The descriptive statistics after organizing the subsamples according to the level of market competition are as follows. The mean (median) cash effective tax rate (CASH_ETR) is −0.224 (−0.216), the mean (median) effective tax rate (GAAP_ETR) is −0.223 (−0.216), and the mean (median) market share (MKTShare) is 0.057 (0.008) in a market with low competition. In addition, in a moderately competitive market, the mean (median) cash effective tax rate (CASH_ETR) is −0.219 (−0.203), the mean (median) effective tax rate (GAAP_ETR) is −0.216 (−0.203), and the mean (median) market share (MKTShare) is 0.040 (0.011). Finally, in a perfectly competitive market, the mean (median) cash effective tax rate (CASH_ETR) is −0.226 (−0.209), the mean (median) effective tax rate (GAAP_ETR) is −0.227 (−0.210), and the mean (median) market share (MKTShare) is 0.027 (0.012).

**Table 4.** Descriptive statistics.

| Variable | Full Samples | | (1) Low Competition | | (2) Mid Competition | | (3) High Competition | |
|---|---|---|---|---|---|---|---|---|
| | Mean (Median) | Max (Min) | Mean (Median) | Max (Min) | Mean (Median) | Max (Min) | Mean (Median) | Max (Min) |
| Cash_ETR | −0.223 (−0.211) | −0.993 (3.070) | −0.224 (−0.216) | −0.993 (3.070) | −0.219 (−0.203) | −0.964 (0.481) | −0.226 (−0.209) | −0.936 (0.197) |
| GAAP_ETR | −0.222 (−0.211) | −0.993 (3.070) | −0.223 (−0.216) | −0.993 (3.070) | −0.216 (−0.203) | −0.964 (0.481) | −0.227 (−0.210) | −0.936 (0.197) |
| MKTshare | 0.045 (0.010) | 0.001 (0.836) | 0.057 (0.008) | 0.001 (0.836) | 0.040 (0.011) | 0.001 (0.384) | 0.027 (0.012) | 0.001 (0.218) |
| Size | 26.209 (25.951) | 23.134 (33.020) | 26.424 (26.029) | 23.134 (33.020) | 26.136 (26.022) | 23.452 (30.933) | 25.860 (25.803) | 23.506 (29.682) |
| Leverage | 0.381 (0.360) | 0.001 (1.923) | 0.391 (0.370) | 0.001 (1.691) | 0.386 (0.376) | 0.001 (1.923) | 0.357 (0.314) | 0.024 (1.535) |
| CurrRatio | 0.530 (0.511) | 0.007 (3.459) | 0.517 (0.502) | 0.015 (3.294) | 0.502 (0.483) | 0.007 (3.459) | 0.587 (0.559) | 0.007 (2.484) |
| lnAGE | 3.321 (3.367) | 2.079 (4.795) | 3.314 (3.367) | 2.079 (4.584) | 3.362 (3.433) | 2.079 (4.762) | 3.290 (3.295) | 2.079 (4.795) |

**Table 4.** *Cont.*

| Variable | Full Samples | | (1) Low Competition | | (2) Mid Competition | | (3) High Competition | |
| --- | --- | --- | --- | --- | --- | --- | --- | --- |
| | Mean (Median) | Max (Min) | Mean (Median) | Max (Min) | Mean (Median) | Max (Min) | Mean (Median) | Max (Min) |
| ROA | 0.062 (0.046) | −0.024 (1.832) | 0.057 (0.043) | −0.024 (0.993) | 0.060 (0.046) | 0.001 (1.832) | 0.072 (0.058) | −0.015 (0.693) |
| MTB | 1.439 (1.023) | 0.172 (26.827) | 1.264 (0.960) | 0.172 (13.284) | 1.488 (0.983) | 0.230 (26.827) | 1.735 (1.280) | 0.301 (16.180) |
| Lossdummy | 0.094 (0.000) | 0.000 (1.000) | 0.104 (0.000) | 0.000 (1.000) | 0.082 (0.000) | 0.000 (1.000) | 0.089 (0.000) | 0.000 (1.000) |
| MKT | 1.560 (2.000) | 1.000 (2.000) | 1.532 (2.000) | 1.000 (2.000) | 1.502 (2.000) | 1.000 (2.000) | 1.683 (2.000) | 1.000 (2.000) |
| n | 4542 | | 2194 | | 1261 | | 1087 | |

Detailed definition of variables is follows. Cash_ETR = cash tax paid divided by before-tax book income; GAAP_ETR = total income tax expense divided by before-tax book income; MKTshare = sales of firm divided by the total sales of the belongs industry; Size = natural logarithm of total assets; Leverage = total liabilities divided by total assets; CurrRatio = current assets divided by underlying assets; MTB = common stock divided by total equity; lnAGE = natural logarithm of current year minus the year of establishment; ROA = net income divided by total assets; Lossdummy = dummy variable that is 1 if loss occurred in the previous year, or 0 otherwise; MKT = 1 if listed in KOSPI, else 2.

Table 5 presents the results of the statistical difference analysis of the mean (median) of the variables categorized by level of competition. It shows that, overall, there are no significant differences in the cash effective tax rate and effective tax rate variables based on the level of competition in the market. In contrast, there are significant differences in the mean (median) of most of the control variables by market competition level.

**Table 5.** Results of Difference test.

| Variable | (1) Low Sample | (2) Mid Sample | (3) High Sample | t−Stat [z−Stat] (1)–(2) | t−Stat [z−Stat] (2)–(3) | t−Stat [z−Stat] (1)–(3) |
| --- | --- | --- | --- | --- | --- | --- |
| | Mean (Median) | Mean (Median) | Mean (Median) | | | |
| Cash_ETR | −0.224 (−0.216) | −0.219 (−0.203) | −0.226 (−0.209) | −0.91 [−3.24 ***] | 1.26 [1.57] | 0.47 [1.29] |
| GAAP_ETR | −0.223 (−0.216) | −0.216 (−0.203) | −0.227 (−0.210) | −1.45 [−3.68 ***] | 1.92 * [1.93 *] | 0.63 [1.28] |
| MKTshare | 0.057 (0.008) | 0.040 (0.011) | 0.027 (0.012) | 4.39 *** [5.34 ***] | 5.74 *** [1.78 *] | 7.65 *** [7.32 ***] |
| Size | 26.424 (26.029) | 26.136 (26.022) | 25.860 (25.803) | 5.56 *** [3.07 ***] | 6.26 *** [4.71 ***] | 10.85 *** [7.64 ***] |
| Leverage | 0.391 (0.370) | 0.386 (0.376) | 0.357 (0.314) | 0.67 [0.42] | 3.31 *** [3.97 ***] | 4.27 *** [4.52 ***] |
| CurrRatio | 0.517 (0.502) | 0.502 (0.483) | 0.587 (0.559) | 1.77 * [2.16 **] | −8.33 *** [−9.30 ***] | −7.78 *** [−7.55 ***] |
| lnAGE | 3.314 (3.367) | 3.362 (3.433) | 3.290 (3.295) | −2.63 *** [−2.94 ***] | 3.28 *** [3.79 ***] | 1.23 [1.60] |
| ROA | 0.057 (0.043) | 0.060 (0.046) | 0.072 (0.058) | −1.20 [−1.52] | −3.89 *** [−5.22 ***] | −6.19 *** [−7.02 ***] |
| MTB | 1.264 (0.960) | 1.488 (0.983) | 1.735 (1.280) | −4.68 *** [−1.89 *] | −3.68 *** [−7.86 ***] | −10.33 *** [−10.90 ***] |

**Table 5.** *Cont.*

| Variable | (1) Low Sample | (2) Mid Sample | (3) High Sample | t−Stat [z−Stat] (1)–(2) | t−Stat [z−Stat] (2)–(3) | t−Stat [z−Stat] (1)–(3) |
| --- | --- | --- | --- | --- | --- | --- |
| | Mean (Median) | Mean (Median) | Mean (Median) | | | |
| *Lossdummy* | 0.104 (0.000) | 0.082 (0.000) | 0.089 (0.000) | 2.10 ** [2.09 **] | −0.58 [−0.58] | 1.36 [1.36] |
| *MKT* | 1.532 (2.000) | 1.502 (2.000) | 1.683 (2.000) | 1.70 * [1.70 *] | −9.02 *** [−8.88 ***] | −8.32 *** [−8.23] |
| n | 2194 | 1261 | 1087 | | | |

The t stat is test statistic of difference analysis for the mean between the two groups, and [z stat] is a test statistic for the Wilcoxon's signed ranks test. *, **, and *** denote the significance at 10%, 5%, and 1% level, respectively. Detailed definition of variables is in Table 4.

Table 6 shows the correlations between the variables used in this study. The cash effective tax rate and effective tax rate were used as the dependent variables in the analytical model, and their correlation coefficients are quite high and significant. This indicates consistency between the measures of tax avoidance. Regarding the correlations of the other variables, the overall correlation coefficients were low, so the concern of multicollinearity was expected to be negligible. Nevertheless, we calculated the variance inflation factor (VIF) for each model and, as expected, the likelihood of multicollinearity was low.

**Table 6.** Correlation matrix.

| Variable | (1) | (2) | (3) | (4) | (5) | (6) | (7) | (8) | (9) | (10) | (11) |
| --- | --- | --- | --- | --- | --- | --- | --- | --- | --- | --- | --- |
| (1) *Cash_ETR* | 1.00 | | | | | | | | | | |
| (2) *GAAP_ETR* | 0.96 | 1.00 | | | | | | | | | |
| (3) *MKTshare* | −0.06 | −0.06 | 1.00 | | | | | | | | |
| (4) *Size* | −0.12 | −0.13 | 0.67 | 1.00 | | | | | | | |
| (5) *Leverage* | −0.10 | −0.11 | 0.11 | 0.20 | 1.00 | | | | | | |
| (6) *CurrRatio* | 0.03 | 0.03 | −0.08 | −0.25 | 0.13 | 1.00 | | | | | |
| (7) *lnAGE* | −0.08 | −0.08 | 0.03 | 0.25 | −0.01 | −0.21 | 1.00 | | | | |
| (8) *ROA* | 0.16 | 0.16 | 0.03 | −0.02 | −0.05 | 0.32 | −0.15 | 1.00 | | | |
| (9) *MTB* | 0.02 | 0.01 | 0.06 | 0.01 | 0.10 | 0.10 | −0.15 | 0.25 | 1.00 | | |
| (10) *Lossdummy* | −0.05 | −0.06 | −0.02 | −0.04 | 0.16 | −0.04 | −0.01 | −0.08 | 0.01 | 1.00 | |
| (11) *MKT* | 0.09 | 0.09 | −0.29 | −0.53 | −0.06 | 0.23 | −0.40 | 0.09 | 0.07 | 0.03 | 1.00 |

Bold denote the significance at 1% level. Detailed definition of variables is in Table 4.

### 4.2. Multivariate Analysis Results

Table 7 presents the analysis results for H1, which states that the level of tax avoidance varies as a company's market power increases. According to the analysis, the coefficients of the market share (MKTShare) variable are 0.0756 (t = 2.39) and 0.0755 (t = 2.45), respectively, in the models with the cash effective tax rate (Cash_ETR) and the effective tax rate (GAAP_ETR) as the dependent variables. Both are significant at the 5% level. This implies that a higher market share is associated with higher levels of tax avoidance. Therefore, H1 of the study is supported. The results of our study are consistent with those of Kim and Lee [6] and Karamshahi et al. [46].

Next, Table 8 presents the analysis results for H2, which states that the level of competition in the market influences the relation between the company's market power and tax avoidance. First, we analyzed the results for the sample with a low level of market competition (monopolistic market). The coefficient of the market share variable (MKTShare) was 0.0696, significant at the 10% level, in the model with the cash effective tax rate (Cash_ETR) as the dependent variable, while it was 0.0818, significant at the 5% level, in the model with the effective tax rate (GAAP_ETR) as the dependent variable. This means that companies in monopolistic markets with low levels of competition engage in more tax avoidance as their market share increases.

**Table 7.** Analysis results.

| Variable | Dependent Variable | |
|---|---|---|
| | *Cash_ETR* | *GAAP_ETR* |
| intercept | 0.1565 (2.08 **) | 0.1546 (2.12 **) |
| MKTShare | 0.0756 (2.39 **) | 0.0755 (2.45 **) |
| Size | −0.0150 (−5.56 ***) | −0.0151 (−5.77 ***) |
| Leverage | −0.0412 (−3.79 ***) | −0.0402 (−3.81 ***) |
| CurrRatio | −0.0224 (−2.21 **) | −0.0236 (−2.39 **) |
| lnAGE | −0.0058 (−1.22) | −0.0046 (−1.00) |
| ROA | 0.3193 (9.45 ***) | 0.3153 (9.60 ***) |
| MTB | −0.0003 (−0.21) | −0.0008 (−0.54) |
| LossDummy | −0.0220 (−3.01 ***) | −0.0228 (−3.22 ***) |
| MKT | 0.0047 (0.87) | 0.0047 (0.88) |
| Year | included | |
| Industry | included | |
| F-value | 8.24 | 8.43 |
| adj. $R^2$ | 0.0683 | 0.0700 |
| Obs. | 4542 | |

**, and *** denote the significance at 5%, and 1% level, respectively. Detailed definition of variables is in Table 4.

**Table 8.** Analysis results.

| Variable | Low Competition | | Moderate Competition | | High Competition | |
|---|---|---|---|---|---|---|
| | Dependent Variable | | | | | |
| | Cash_ETR | GAAP_ETR | Cash_ETR | GAAP_ETR | Cash_ETR | GAAP_ETR |
| intercept | 0.1456 (1.36) | 0.1582 (1.51) | −0.0453 (−0.33) | −0.0511 (−0.39) | 0.6075 (2.89 ***) | 0.5325 (2.57 **) |
| MKTShare | 0.0696 (1.72 *) | 0.0818 (2.08 **) | 0.0675 (0.90) | 0.0546 (0.76) | 0.0511 (0.25) | −0.0078 (−0.04) |
| Size | −0.0152 (−3.95 ***) | −0.0164 (−4.36 ***) | −0.0087 (−1.78 *) | −0.0077 (−1.66 *) | −0.0288 (−3.70 ***) | −0.0263 (−3.42 ***) |
| Leverage | −0.0353 (−2.06 **) | −0.0311 (−1.86 *) | −0.0407 (−2.17 **) | −0.0409 (−2.29 **) | −0.0295 (−1.34) | −0.0318 (−1.46) |
| CurrRatio | −0.0076 (−0.48) | −0.0081 (−0.52) | −0.0043 (−0.25) | −0.0094 (−0.56) | −0.0628 (−3.15 ***) | −0.0610 (−3.10 ***) |
| lnAGE | −0.0013 (−0.18) | 0.0017 (0.24) | −0.0037 (−0.47) | −0.0065 (−0.87) | −0.0183 (−1.82 *) | −0.0150 (−1.51) |
| ROA | 0.3132 (5.52 ***) | 0.3006 (5.44 ***) | 0.2504 (5.06 ***) | 0.2429 (5.15 ***) | 0.4781 (6.28 ***) | 0.4852 (6.45 ***) |
| MTB | 0.0044 (1.37) | 0.0048 (1.51) | 0.0003 (0.14) | −0.0006 (−0.30) | −0.0067 (−2.08 **) | −0.0077 (−2.40 **) |
| LossDummy | −0.0221 (−2.08 **) | −0.0237 (−2.29 **) | −0.0164 (−1.23) | −0.0173 (−1.36) | −0.0318 (−2.11 **) | −0.0309 (−2.08 **) |
| MKT | 0.0002 (0.03) | 0.0030 (0.37) | 0.0191 (2.03 **) | 0.0164 (1.83 *) | −0.0128 (−1.10) | −0.0139 (−1.21) |
| Year | included | | included | | included | |
| Industry | included | | included | | included | |
| F-value | 4.21 | 4.37 | 5.18 | 4.94 | 6.08 | 6.10 |
| adj. $R^2$ | 0.0593 | 0.0620 | 0.0877 | 0.0832 | 0.0972 | 0.0974 |
| Obs. | 2194 | | 1261 | | 1087 | |

*, **, and *** denote the significance at 10%, 5%, and 1% level, respectively. Detailed definition of variables is in Table 4.

On the other hand, in the analysis involving a market with moderate competition, the coefficients of the market share variable (MKTShare) were not significant (0.0675 in the model with cash effective tax rate (Cash_ETR) as the dependent variable and 0.0546 in the model with effective tax rate (GAAP_ETR) as the dependent variable). This suggests that in a moderately competitive market, an increase in an individual company's market share does not increase tax avoidance significantly.

Lastly, the coefficients of the market share variable (MKTShare) in the analysis with the perfectly competitive market were not significant (0.0511 in the model with cash effective tax rate (Cash_ETR) as the dependent variable and $-0.0078$ in the model with effective tax rate (GAAP_ETR) as the dependent variable). This implies that, in a highly competitive market, an increase in the market share of an individual company does not lead to a significant increase in tax avoidance.

The analysis results can be summed up as follows: when the level of market competition is disregarded, tax avoidance increases significantly as a company's market share increases. However, when the sample is subdivided and further analyzed considering the level of market competition, the results show that tax avoidance increases significantly as a company's market share increases only in oligopolistic markets with low market competition; the relationship between market share and tax avoidance is insignificant in other types of markets. This can be interpreted to mean that leading companies in oligopolistic markets can maintain their sustainability even if tax avoidance increases the likelihood of future tax risk, which is not the case for companies in highly competitive industries. Overall, our results suggest that H1, which states that the level of tax avoidance varies as a company's market power increases, mainly holds true for less competitive oligopolistic markets. Therefore, H2, which states that the level of competition in the market influences the effect of a company's market power on tax avoidance, is supported.

## 5. Conclusions

We analyzed the effect of a company's sustainable competitive advantage on its tax avoidance strategy and investigated whether the effect differs according to the level of competition in the company's relevant market.

The result of this study can be summarized as follows. tax avoidance increases significantly as a company's market share increases in full samples. These results of our study are consistent with prior research [6,46]. However, if the sample is divided by the level of market competition and analyzed, the results show that tax avoidance increases significantly with the increase in a company's market power only in oligopolistic markets with low market competition; the relationship between market share and tax avoidance is not significant in other types of markets. Apparently, tax avoidance increases significantly as a company's market share increases only in oligopolistic markets with low market competition; the relationship between market share and tax avoidance is insignificant in other types of markets. Therefore, it is interpreted that the relationship between the competitive advantage of companies and their tax avoidance varies depending on the level of competition in the market. These results of our study have not been studied in previous studies. While previous studies analyzed only the impact of individual companies' competitive advantage on tax avoidance, this study is significant as it provides empirical results that also consider the level of competition in the market. However, this study has a limitation in that it analyzed only domestic companies. If the model of this study is empirically analyzed for companies in different countries, the results of our study may differ. Therefore, in the future, our researchers will construct a multinational sample and empirically analyze the models of this study to solidify the subject of this study.

**Author Contributions:** Conceptualization, Y.S. and J.-M.P.; formal analysis, Y.S.; methodology, Y.S.; visualization, J.-M.P.; writing—original draft, Y.S. and J.-M.P.; writing—review and editing, Y.S. and J.-M.P. All authors have read and agreed to the published version of the manuscript.

**Funding:** This research received no external funding.

**Institutional Review Board Statement:** Not applicable.

**Informed Consent Statement:** Not applicable.

**Data Availability Statement:** All data that can reproduce the results in this study can be requested from the corresponding author.

**Conflicts of Interest:** The authors declare no conflict of interest.

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
