# Peer review of "The Effect of a Company’s Sustainable Competitive Advantage on Their Tax Avoidance Strategy—Focusing on Market Competition in Korea"

_sustainability, doi:10.3390/su15107810_

Round 1
Reviewer 1 Report
I have two remarks:
1. The title should be improved (why "focusing on market strategy"?). The Korean case should be mentioned.
2. The behavioural aspects of TMT decisions should be mentioned. The prospect theory by Tversky and Kahneman can be supportive.
Author Response
We appreciate the time and effort that you and the reviewers dedicated to providing feedback on our manuscript and are grateful for the insightful comments on and valuable improvements to our paper. We have reflected most of the suggestions made by the reviewers. Please refer to the attached file for details.

Reviewer 2 Report
Following are my comments.
1. Introduction:
Introduction
The introduction is about the hook/gap of the study. I am unable to understand the gap in the
literature. The paper contributions are unclear, and why this study is required. How is it
different from previously available studies?
Literature review
In general, the section provides adequate literature in explaining the relationship between the
study constructs. However, the arguments leading to mediated hypothesis are thin and need
to be strengthened. In addition, I think it is not enough to provide one paragraph. I think the proper theory is also required in developing the hypothesis.
Methods The method does not provide the data collection procedure.
· Why this context is important? Why was this study population targeted?
· When was the data collection carried out?
· Which sampling technique was used?
· Which method was used for calculating the sample size?
· Please clearly explain how the adapted scales are appropriate for this research.
Results:
· The analysis techniques are appropriate, and the results are adequately explained.
Conclusion and implications
· It is difficult to understand the novel implications of the study.
Additional comments
· Due to grammatical errors and typos, proofreading is required.
· Due to grammatical errors and typos, proofreading is required.
Author Response

(The authors gave the same response as above.)

Reviewer 3 Report
This article analyzes whether a company's competitive advantage affects a company's tax avoidance strategy. Also, it analyzes whether these effects depend on the level of competition in the market to which the company belongs.
Strong points of the article:
1. The topic of tax avoidance is very important and deserves to be addressed with good research. This manuscript does that.
2. The article is well organized and follows the proper methodology.
3. The results are accompanied with strong empirical analysis results.
Following are the weak points those need to be addressed:
1. Existing research gaps on tax avoidance should be more emphasized in the introduction.
2. What are the limitations of the proposed mathematical model?
3. Explain clearly how equation (2) was defined. Is it a new contribution by the authors or an extension of an existing equation. Provide proper reference.
Minor spell check required.
Author Response

(The authors gave the same response as above.)

Reviewer 4 Report
Dear Authors,
I’ve read your paper and I have some suggestion, before the editor could proceed with the publication.
1. The introduction opens with the aim of the paper, but it should open with the framework of the topic and, only in the end, your purpose.
2. The literature review must be reorganized, even with the introduction of a summary table, in order to underline the literature gap, you want to cover.
3. You often write “as discussed in many previous study”, but you don’t provide any reference, please fix it.
4. In the conclusion, you should recall the literature gap and provide secondary literature; limitations of the study and emerging issues are missing.
Author Response

(The authors gave the same response as above.)
